



# Simulating Synthetic Tropical Cyclone Tracks for Statistically Reliable Wind and Pressure Estimations

Kees Nederhoff[1], Jasper Hoek[2,3], Tim Leijnse[2], Maarten van Ormondt[2], Sofia Caires[2], Alessio Giardino[2]

[1] Deltares USA, 8601 Georgia Ave, Silver Spring, MD 20910, USA
[2] Deltares, Boussinesqweg 1, Delft, 2629 HV , The Netherlands
[3] Delft University of Technology, Stevinweg 1, 2628 CN Delft, Netherlands

*Correspondence to*: Kees Nederhoff (kees.nederhoff@deltares-usa.us)

**Abstract.** The design of coastal protection measures and the quantification of coastal risks at locations affected by tropical
cyclone (TC) are often based solely on the analysis of historical cyclone tracks. Due to data scarcity and the random nature of TCs, the assumption that future TCs could hit a neighboring area with equal likelihood than past events can potentially lead to over- and/or underestimations of extremes and associated risks. The simulation of numerous synthetic TC tracks based on (historical) data can overcome this limitation. In this paper, a new method for the generation of synthetic TC tracks is proposed. The method has been implemented in the highly flexible open-source Tropical Cyclone Wind Statistical Estimation Tool
(TCWiSE). TCWiSE uses an Empirical Track Model based on Markov-chains and can simulate thousands of synthetic TC tracks and wind fields in any oceanic basin based on any (historical) data source. Moreover, the tool can be used to determine the wind extremes and the output can be used for the reliable assessment of coastal hazards. Validation results for the Gulf of Mexico show that TC patterns and extreme wind speeds are well reproduced by TCWiSE.

## 1 Introduction

Tropical Cyclones (TCs) are among the most destructive natural hazards worldwide. TCs can cause hazardous weather conditions including extreme rainfall and wind speeds, leading to coastal hazards such as extreme storm surge levels and wave conditions. In multi-hazard risk assessments, the spatial distribution of surface winds is needed. Past observed best-track data (BTD) can be used to reliably reproduce spatially varying wind conditions during individual TCs using parametric models (Nederhoff et al., 2019) and consequent hazards (e.g. Giardino et al., 2019). However, BTD covers only historical occurrences,
readily available from the 1970s onward globally, which in many regions implies that no reliable hazard estimates can be derived solely from the BTD fields due to a lack of occurrences in this (relatively limited) timespan.

      Extreme value theory is concerned with the distribution of rare events, rather than usual occurrences. For TCs, one often refers to either the first-order hazards due to the TC (e.g. maximum wind speed), or to second-order effects (e.g. storm surge levels and wave heights). This is required for example to define design conditions for coastal protection measures or to
quantify coastal risks. A wide range of such statistical methods exists for all of which it is of importance to use numerous observational points to derive reliable extreme values. When the dataset covers the return period of the event, then the extreme





value estimation can be based directly on historical values (i.e. non-parametric). However, for the estimation of extremes associated with longer return periods, one needs to resort to the fitting of a statistical distribution to the data (i.e. parametric). The simplest technique is either to fit a Gumbel distribution with two parameters (location and scale) under certain assumptions or to fit a Generalized Extreme Value (GEV) to a time series of annual maxima (Coles, 2001). Other methods make better use

of the available data, for example, via a peaks-over-threshold (POT) approach to identify all extremes within a year and to fit the Generalized Pareto distribution (GPD) to them (Caires, 2016).

Worldwide the length of TC track records varies from approximately 50 years (from 1970's onward) to more than 150 years in the Gulf of Mexico (GoM). Thus, depending on the region, the number of events recorded in the direct vicinity of a location varies significantly. Furthermore, in certain regions, the frequency of occurrence is also very low, making the

sample size of historical events very limited. Only using a handful of observed TCs in recent history has severe limitations when estimating extreme wind, storm surge and wave conditions for rare (e.g. 1 000 years) return periods, since individual storms will start to affect the derived extremes. In particular, biases (both over- and underestimations) start to emerge due to sampling errors.

To overcome this data scarcity problem, one potential approach is to generate synthetic TC tracks, which increases

the amount of data by introducing cyclones that could potentially occur. Two different types of models are available for the generation of synthetic tropical cyclones. These are the Simple Track Model (STM) and the Empirical Track Model (ETM). STM (e.g. Vickery & Twisdale, 1995) was the first method developed to generate synthetic cyclones. The basic idea is that specific observed TC characteristics (e.g. wind speed, central pressure deficit, the radius of maximum winds (RMW), heading, translation speed, coast crossing position, etc.) are obtained and used to construct probability density functions. Next, these

characteristics are sampled from their distributions using Monte Carlo simulations and passed along a track that does not vary. This means that TC characteristics are kept constant along the track. The downside of this method is that it is very site-specific as all parameters are gathered for a single area or coastline. ETM is in principle the evolution of STM (e.g. Vickery et al., 2000). It uses the same technique of gathering the statistics and then sampling them, utilizing Monte Carlo simulations. Instead of sampling all parameters once, the characteristics can change in its characteristics every time step along the track.

In recent literature, several synthetic TC databases and or methods have been published. Vickery et al. (2000) used statistical properties of historical tracks and intensities to generate a large number of synthetic storms in the North Atlantic basin. Six-hour changes in TC properties were modeled as linear functions of previous values of those quantities as well as position and sea surface temperature. James and Mason (2005) applied a similar, yet slightly simpler and less data-intensive, approach since the focus was on synthetic TCs affecting the Queensland coast of northeastern Australia where fewer data was

available compared to the North Atlantic basin. Vickery et al. (2009) added a second step in the TC generation by including thermodynamic and atmospheric environmental variables such as sea surface temperature, tropopause temperature and vertical wind shear. Emanuel et al. (2006) also used the ETM; however, for the generation of the synthetic tracks they applied Markov chains (Brzeźniak & Zastawniak, 2000) with kernel density estimates (KDE) conditioned on a prior state, time, and position,





instead of using a linear function. Bloemendaal et al. (2020) developed a synthetic TC database on a global scale following the principles outlined in James and Mason (2005).

While there are numerous methods and tools available to generate synthetic TCs, most of them were developed with a very specific focus in mind and therefore not always easily applicable to other areas in the world and/or different applications.

Moreover, none of these methods are yet available open-source for review by other peers and potentially peer-application and all are focused purely on the generation of the track itself. For example, for coastal engineering or risk-based applications, the possibility to easily link the track to other processes (e.g. generation of wind profiles, rainfall, hazard modeling) could offer a wide range of opportunities for different applications.

In this paper, a new method for the generation of synthetic cyclone tracks and wind fields is described. The method

has been implemented in a new tool to compute synthetic TC tracks, based on the ETM method, for any oceanic basin in the world. This new tool, named TCWISE (Tropical Cyclone Wind Statistical Estimation Tool), has been made publicly free and open-source via URL (will be made available after acceptance of the paper). The tool is set up as a Markov model where (historical) meteorological data serves as a source to compute synthetic tracks. Additionally, TCWISE can create meteorological forcings for further use in different hazard models (e.g. surface wind fields, TC induces rainfall, etc.) including

the possibility to assess current and future climate variability.

TCWiSE has been developed with an attempt to give users flexibility in its choices. For example, while a comprehensive historical TC database is already included in IBTrACS (International Best Track Archive for Climate Stewardship, Knapp et al., 2010), the tool offers the option to choose from different sources within this dataset. Additionally, variables like the resolution of kernel density estimation (KDE) and internal parameters can be tuned if wanted. Also, it is

possible to choose among several wind profiles to create spatially varying wind fields. This approach makes it possible to calibrate parameters in TCWiSE that arguably vary from case study to case study. TCWiSE has been successfully applied in several studies prior to this publication (e.g. Deltares, 2016, Hoek, 2017 and Bader, 2019). In general, the whole tool is data-driven but due to the usage of Markov-chains and KDE, variability within the dataset can also be explored (i.e. combinations of statistical plausible parameters that have not occurred historically).

This paper is outlined as follows: Section 2 describes the method and code structure of TCWiSE. Section 3 presents a validation case study for the GoM. Finally, Section 4 and 5 discuss and summarize the main conclusions of the study.

## 2. The Synthetic Track Generation with TCWiSE

### 2.1 Introduction

TCWiSE comprises a Monte Carlo method for synthetic TC generation and involves four main steps: track initiation, track

evolution, wind field construction and determination of extreme surface wind speeds. The steps are: 1) Based on the average number of TCs per year, its monthly distribution and the distribution of the genesis location, timestamps and synthetic genesis locations are generated. 2) Subsequently, an ETM is used to determine the changes in track and intensity at certain time





intervals (i.e. 3-hourly per default). The ETM is a Markov model where the values of the next time step solely depend on that of the previous time step, similar to the methods developed by e.g. Vickery et al. (2010) and Emanuel et al. (2006). The main variables it keeps track of are location (latitude and longitude), time, maximum sustained wind speeds ($v_{max}$), forward speed (c) and heading (θ) of the synthetic TC track. 3) After the TC tracks have been generated, the surface wind fields are constructed

using the so-called updated Holland wind profile (Holland et al., 2010) using calibrations based on Nederhoff et al. (2019). 4) Finally, the generated data of surface wind fields is used to estimates TC wind extremes. The main outputs of the tool are the synthetic tracks, the surface wind fields per TC and the surface wind extremes. The output wind fields can be used further to derive extremes of associated second-order effects, such are storm surges and waves. The tool is written in Matlab and leverages the Parallel Computing Toolbox to allow the utilization of the multicore processors on computer clusters.

**2.2 Flowchart**

A compact flowchart of the method which is used to generate the synthetic tracks is shown in Figure 1.

The steps are as follows:

1.  **Define settings**: The user specifies the data source, the area of interest, the number of years which are to be simulated

and a number of numerical parameters. In particular, the included IBTrACS dataset contains data from several meteorological agencies from which the user can choose. Also, the users can define settings such as the number of points needed per KDE. The user can also define bulk climate variability parameters such as changes in TC frequency and intensity due to climate change.

2.  **Construct statistics of original data**: TCWiSE processes the (historical) data and computes the probability of genesis

and termination per location on the map. Moreover, it computes change functions for the three variables the tool keeps track of. In particular, KDE of the conditional-dependent changes in maximum sustained wind speeds (intensity; $v_{max}$), forward speed (c) and heading (θ) as a function of the location and the variable itself are determined and saved for later usage. This information will be used within the Markov-chains during simulation of synthetic tracks.

3.  **Compute cyclone genesis**: The tool computes the number of storms to be generated by taking the average number of

storms observed per year within the oceanic basin of interest. The yearly distribution (i.e. seasonality) is also taken into account by using a Poisson distribution, giving each track a unique timestamp within the number of years to be simulated. For every track, its genesis location is determined and each TC track gets an associated initial $v_{max}$, c, and θ associated to the genesis location. See section 2.3 for more information.

4.  **Compute new location and intensity**: For every track, TCWiSE samples on 3-hourly intervals change to the three

sampled parameters ($v_{max}$, c, and θ) until termination of the track. KDE is used to randomly sample changes to these parameters as a function of location and the parameter itself. The tool uses the maximum sustained wind speeds as the intensity parameter. Heading and forward speed are the location parameters. All these three parameters are sampled at a use-definable time step (3-hourly by default).



5. **If on land (optional):** It is possible to include an additional decrease in intensity on land via relationships developed by Kaplan and DeMaria (1995). Implicitly, part of the decrease of intensity on land is already accounted for via the KDE of $v_{max}$. However, due to search windows, some of this effect is smoothed out. See section 2.4.2 for more information.

6. **Terminate track:** After each interval of 3 hours, the tool checks if the tracks should be terminated. The termination criteria are defined in three different ways: a. probability, b. wind speed criteria, c. sea surface temperature (SST). See section 2.4.3 for more information.

7. **Valid track:** To make sure realistic TC tracks are generated, the tool checks if the synthetic track that is terminated has reached a wind speed of at least 25 m/s or more during its lifetime (i.e. approximate TC category 1 based on Saffir-Simpson Hurricane Wind Scale). This prevents the generation of extratropical storms that never reach TC status.

8. **Last track:** TCWiSE continues with this loop until the last synthetic TC track has been generated.

9. **Create spatially-varying wind field maps**: The tool creates meteorological forcing conditions, i.e. the surface wind fields, for subsequent analysis of wind swath maps and for the application within numerical models (currently only Delft3D4 and Delft3D-FM are supported including flow and wave; Lesser et al., 2004 and Kernkamp et al., 2011).

10. **Create wind swaths**. TCWiSE creates wind swaths (footprints or extreme surface wind velocities) of the high TC winds, based on the wind field maps. In particular, output wind speeds are in m/s and, by default but user-definable, 10-minute averaged. Note that different meteorological agencies use different wind speed averaging periods. Harper et al. (2010) recommend for at-sea conditions a conversion factor of 1.05 going from 10-minute to 1-minute averaged wind speeds.

A more detailed description of the track initialization, track & intensity evolution, termination, climate variability and wind fields is described in the paragraphs below.



**Figure 1. Flowchart of the track modeling procedure. Dark blue colors are pre-processing steps, blue colors the computational core of TCWiSE and light blue post-processing steps. KDE stands for Kernel Density Estimation, SST for Sea Surface Temperature and POT/GPD is the acronym of Peak Over Threshold and Generalized Pareto Distribution.**





### 2.3 Track initialization

The track initialization is done through random sampling of the genesis locations for each track from a spatially-varying probability constructed based on (historical) input data. Only the spatial occurrence of the genesis locations is sampled, as no temporal variability of genesis locations or other input parameters are included in the tool. The spatial-varying probability used

to sample the genesis locations is constructed by first drawing a rectangular grid of a user-definable size (default: 1 x 1°) around all historical events under consideration. For each grid point, all genesis locations within a certain distance (default 200 km in size, but user-definable) distance are counted and normalized with the total number of counted genesis points to obtain the genesis density at each grid point.

      Genesis location in ocean surface temperatures less than a user-definable value (default: 24 degrees Celsius), are

deleted since high SST is the driving force behind TC genesis and without it, TCs cannot occur (e.g. Gray, 1968). TCWiSE uses SST data from the International Research Institute of Columbia University (2017) which provides a worldwide monthly average SST map on a 1-degree resolution.

      After generating the genesis locations, the matching intensity and track propagation of genesis are determined. The intensity, heading and forward speed of the TC at the genesis location are determined by randomly sampling from all the

historical occurrences at genesis again within a certain distance user-definable size. Hence, this sampling is using only the data points during TC genesis and results in initial values for intensity, heading and forward speed (and not changes to these variables as will be done for the track evolution).

### 2.4 Track evolution

After the generation of the genesis location and parameters, the evolution of the track and intensity is modeled during its

lifetime in (by default) 3-hourly intervals. The propagation is modeled by sampling the change in the heading ($\Delta\theta$), forward speed ($\Delta c$) and intensity ($\Delta v_{max}$) for each time step.

### 2.4.1. Search range

The KDE that is sampled are constructed for each grid point based on input data within a specific search range. This search range is defined by a rectangular box of a user-definable size (default: 1 x 1°) around the point of interest. The minimum

number of data points required within the search range is 250 data points (default, user-definable). If less than the specified amount of points are located within the search range, the search range is increased until the required number of data points are found or until the maximum is reached (user-definable; default 5° × 5°).

      The change in intensity evolution and track propagation, which includes the heading and forward speed, are all treated similarly. Changes are sampled from the pre-processed KDE that are conditionally dependent on the previous time step.

Historical occurrences are smoothed since a kernel density estimation (KDE) from raw histograms (Wand and Jones 1994). This overcomes possible discrete signals. By default, the heading is divided into 17 equally large bins and partly overlapping





bins of 45 degrees, forward speed is divided into 17 overlapping bins of 2.5 knots and wind speed is divided into 37 overlapping bins of 5 knots. This ensures that the full parameter range for TCs is covered. For each variable, the search range (i.e. range for which values are included in the bin) is twice the window size (i.e. the difference per each subsequent bin) to ensure a smooth transition between different bins. All these settings are user-definable. Data points that are on land can be excluded in

the computation of the intensity evolution.

No additional parameters are defined for the track evolution. Effects such as intensification, Coriolis, wind shear, beta drift (Holland, 1983), etc. are not explicitly defined nor controlled for. The conditional-depend KDE of change per variable per location drives the complete track evolution.

### 2.4.2. Effect of land

When a TC makes landfall, TCs weaken due to, among other factors, a lack of heat sources (e.g. Tuleya, 1994). This effect should be part of the conditional-depend KDE, but due to the possibly large search ranges per location (and thus blending of on-land and on-water conditions), the effect of land can be underestimated (and the intensity on-water underestimated). Therefore, the user can exclude data points on land. When this is chosen, one should use additional formulations to reduce intensity when the synthetic TC is on land. Among other relationships available in literature, Kaplan and DeMaria (1995)

created a simple empirical model for computing cyclone wind decay after landfall. In TCWiSE, a similar method can be used to compute the decay of wind speed after landfall. Following the relationships of Kaplan and DeMaria, wind speed decreases exponentially based on how long a TC is on land. The specific amount of decay as a function of time is, again, user-definable.

### 2.4.3. Track termination

During each interval of 3 hours, the tool checks if the tracks should be terminated. The termination criteria is defined in three
different ways:

1. When the wind speed is lower than a user-definable low value (user-definable, default 10 kn/s).
2. When the synthetic TC is over a user-definable low water temperature (user-definable, default 10 degrees Celsius).
3. The probability of termination based on (historical) input data.

When different methods of termination are used, the termination of a synthetic TC is thus not completely similar to the historical probability of termination. Hence, termination within TCWiSE can also be triggered by low wind speeds (due to the fact the TC is on land) and/or too low SST.

### 2.4.4. Climate variability

Projected effects of climate change on frequency and intensity of TCs can also be taken into account via the heuristic
implementation of a factor on both the frequency and intensity. These factors can be defined using expert assessment of TC climate predictions (e.g. Knutson et al. 2015), allowing, for instance, the assessment of changes in TC coastal hazards in the


next century. The effect of climate variability on possible shifts of the TC tracks or regional changes of parameters are not included yet but could be included by modifying the (historical) KDE's and/or using global climate models as an input source for TCWiSE.

### 2.5. Spatially-varying wind field

After the generation of the track (time, location and intensity), spatially varying wind fields are computed based on the parametric model of Holland et al. (2010) via the Wind Enhanced Scheme (WES; Deltares, 2018). The relationships of Nederhoff et al. (2019) are used to compute either the most probable TC geometry (radius of maximum winds; RMW and radius of gale-force winds R35) or to take geometry into account as a stochastic variable. The user has the choice between generic relationships and calibrations for different basins. This ensures reliable azimuthal wind speeds. TC asymmetry is
considered based on Schwerdt (1979) and assumes a constant inflow angle of 22 degrees (Zhang & Uhlhorn, 2012).

### 2.6. Wind swaths

After the generation of the spatially varying wind fields, wind swaths (footprints of extreme surface wind velocities) for different return periods are generated. Both non-parametric as parametric extremes based on a fitted POT/GPD wind swaths for different return periods are computed. TCWiSE utilizes the peaks-over-threshold method (POT) combined with the
Generalized Pareto distribution (GPD; Caires, 2016) for extreme value analysis. In particular, the choice of the threshold for the POT and the fitting of the coefficients are automatically performed.  Parametric estimates of extremes are preferred when 1) statistical uncertainties need to be quantified or 2) when fewer observations are available to base the non-parametric estimates on.

## 3. Tool validation

### 3.1. Introduction

The United States (US) is one of the countries most affected by TCs over the years. Especially the United States Gulf Coast has suffered severely from hurricanes in the past which have caused a significant number of casualties and damage. Among the most sadly famous, TCs Andrew 1992, Katrina 2005 and Harvey, 2012 have devastated US territory. In the severe hurricane season of 2017 alone, Harvey, Irma and Maria resulted in more than 250 billion USD in damage in the US (NOAA, 2018).
In this section, a validation of generation, occurrence, propagation and termination of synthetic TC is carried out, by comparing with historical tracks for the entire North Atlantic (NA) basin. A more detailed comparison between historical BTD from the IBTrACS database and simulated synthetic tracks by TCWiSE is performed for 9 control points in the Gulf of Mexico (GoM). Subsequently, extreme wind speed estimates from TCWiSE and from historical data are compared along the coastline and also validated against literature. Figure 2 presents the area of interest for the validation case study, including relevant
locations for this analysis.





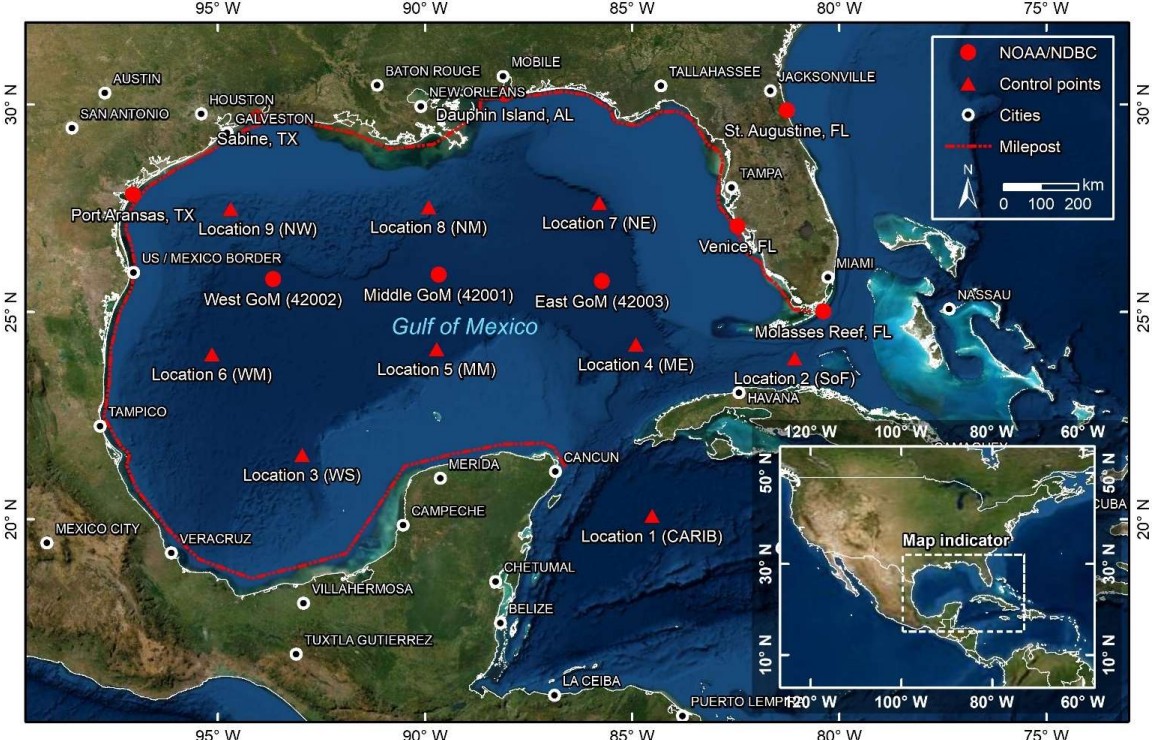

**Figure 2. Area of interest of the Gulf of Mexico, including the locations of the 9 control points in the GoM (red triangles), NOAA/NDBC measurement location (red dots), cities (white circles) and milepost (red dashed line). © Esri, DigitalGlobe, GeoEye, Earthstar Geographics, CNES/Airbus DS, USDA, USGS, AeroGRID, IGN, and the GIS User Community.**



### 3.2 Data

The NA basin data from IBTrACS database are used within the TCWiSE algorithm to compute 10 000 (ten thousand) years of synthetic TCs. Only historical data observed from 1886 up to 2019 are considered. This is because from 1886 the accuracy in measurements of the maximum sustained wind speeds has increased from 10 kt to 5 kt. This yields in 955 historical TCs

and 71 320 synthetic TCs for the entire NA basin.

Measured winds from a total of 9 National Data Buoy Center (NDBC https://www.ndbc.noaa.gov/) buoys across the GoM have been used in this study to validate the TCWiSE computed extreme wind speeds. Computed and observed wind speed are all converted (if needed) to 10-meter height and 10 minute-averaged (see e.g. Harper et al. 2010). This is generally the height and the averaging period needed for hydrodynamic models (in wave modeling this is generally the 1-hour average

wind speed). Only observations from buoys with at least 20 years of data have been used to validate modeled wind speeds. Moreover, only observations within a 200 km radius of an active TC (based on IBTrACS) are considered. This prevents the inclusion of peak wind speeds due to extra-tropical storms instead of TCs in the validation.

Moreover, TCWiSE computed extreme values are compared to values found in the literature. For example, Vickery et al. (2009) present simulated TC induced wind speeds across the US coastline for return periods of 50 up to 2000 years.

Following the methodology of Neumann (1991), along the US coastline, NOAA (National Oceanic and Atmospheric Administration, https://www.noaa.gov/) presents hurricane return periods for both hurricanes (>64 knots) and major hurricanes (>96 knots) within 50 nautical miles (92.6 kilometers) based on the track information. TCWiSE computed return periods are compared to NOAA's reported values (https://www.nhc.noaa.gov/climo/).

### 3.3. Validation computed vs. historic cyclone parameters

### 3.3.1. Statistical test for TC validation

A variety of tests is available for statistical comparison between computed and historical cyclone parameters. The tests are used to prove the hypothesis that the historical values come from the same statistical population as the simulated values. For each parameter, such as forward speed, a goodness of fit between the historical cumulative distribution function (CDF) can be performed and compared to the CDF from the synthetic tracks. Strictly, this would require that different data sets are employed

for model fitting and for model testing so that distributional parameters of the model used to generate the large-sample CDF are not estimated from the historical sample. However, in this paper, we utilized all available observational data to include as much climate variability in the synthetic tracks as possible.

Several tests exist (e.g. Kolmogorov–Smirnov, Cramer-von Mises, Anderson-Darling, Kuiper, Watson) to test the null hypothesis that the samples x and y come from the same (continuous) distribution (Stephens, 1974). In addition, a more

pragmatic approach is available which consists of simply computing the mean-absolute-error (MAE) or regression on the CDFs. In this paper, we present a combination of different statistics to test if the synthetic tracks have similar statistical properties as the Best Track Data (BTD). In particular, normalized mean-absolute error (nMAE), root-mean-square error



(RMSE) and bias are presented. Moreover, the CDF of several TCs physical properties are compared for the historical and synthetic tracks. Finally, the Kirchhofer (1974) method is used for quantifying similarities/differences in spatial patterns (e.g. TC genesis, evolution, etc.).

### 3.3.2 Computed and historic TC parameters

In the following paragraphs, the modeled results of TCWiSE are compared to historical BTD. Validation follows the life of a TC with first a visually and qualitative validation of the generation being presented. Subsequently, the track occurrence, evolution and CDFs of the three main parameters of TCWiSE are compared quantitatively to historical data. Lastly, a visual and qualitative validation of the termination is presented.

*Generation*

Historical and simulated genesis probability for the entire North Atlantic Basin (NA) is shown in Figure 3. Cyclone genesis is taken as the first point which the BTD identifies as such, which means it is the point from where meteorological institutes started tracking the storm. As shown in Figure 3A, visually the simulated and the historical genesis match well. A hot spot of TC genesis is shown on the West coast of the African continent. Additionally, two hot spots are visible east of the Caribbean

Sea and in the western part of the Caribbean. Within the GoM some areas also show cyclone genesis. The genesis patterns are almost identically while being slightly smoothed out in the simulated synthetic tracks (Figure 3B). This visual assessment was quantified and confirmed by using the Kirchhofer metric score, which provided a value equal to 0.967 (a value of 1.0 represents a perfect match). In particular, grid cells that are zero (either in the historical or synthetic dataset) are not taken into account in the analysis. This gives confidence that TCWiSE can reproduce the genesis patterns observed in the historical BTD.

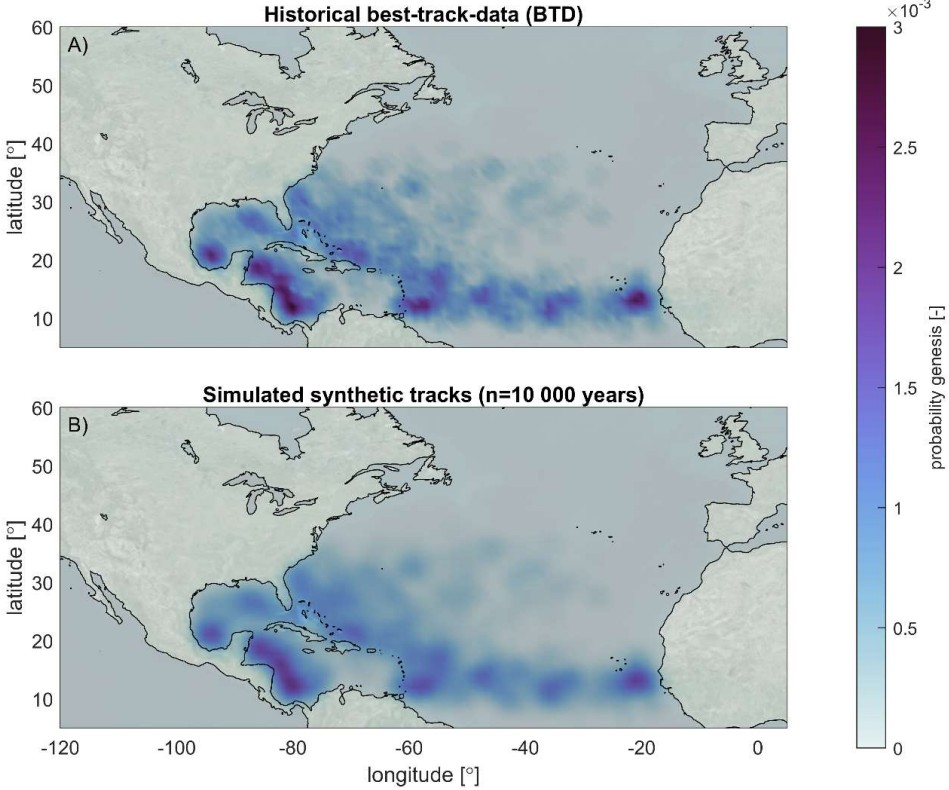

**Figure 3. Genesis probability of historical (BTD) (upper panel; A) and simulated TCs with TCWiSE for the NA basin (bottom panel; B). Occurrence is based on TCs within 200 km per grid cell for historical TCs from 1886-2019 and 10 000 years of simulated events. © Microsoft Bing Maps.**

*Track occurrence, evolution and CDFs*

Historical and simulated TC intensity tracks are shown in Figure 4. All individual tracks are plotted with a color code derived

from the intensity of the eye of the storm (i.e. maximum sustained wind speed). Tracks with higher intensity are plotted on top

of those with lower ones. The figure shows that TCs are generated around latitudes of +/- 10-20 degrees (see also Figure 3).

Part of the TCs increase in intensity while moving towards the north-west making landfall in the US, Central America, northern

countries of South America and across the Caribbean. Others turn back in an eastward direction and propagate towards Europe.

Intensities are generally larger in the Caribbean and GoM while TCs that propagate northward decrease in intensity.  Similar

patterns can be observed in the simulated synthetic TCs (Figure 4B). However, higher intensities can be observed for individual

simulated synthetic tracks due to the larger number of years of data that are presented (10 000 years simulated tracks vs 134

years for the historical tracks) and thus a larger likelihood of having a more intense TC.  Moreover, it does seem that synthetic

TC tracks have a less clear southwest-northeast orientation in heading on the North Atlantic Ocean.

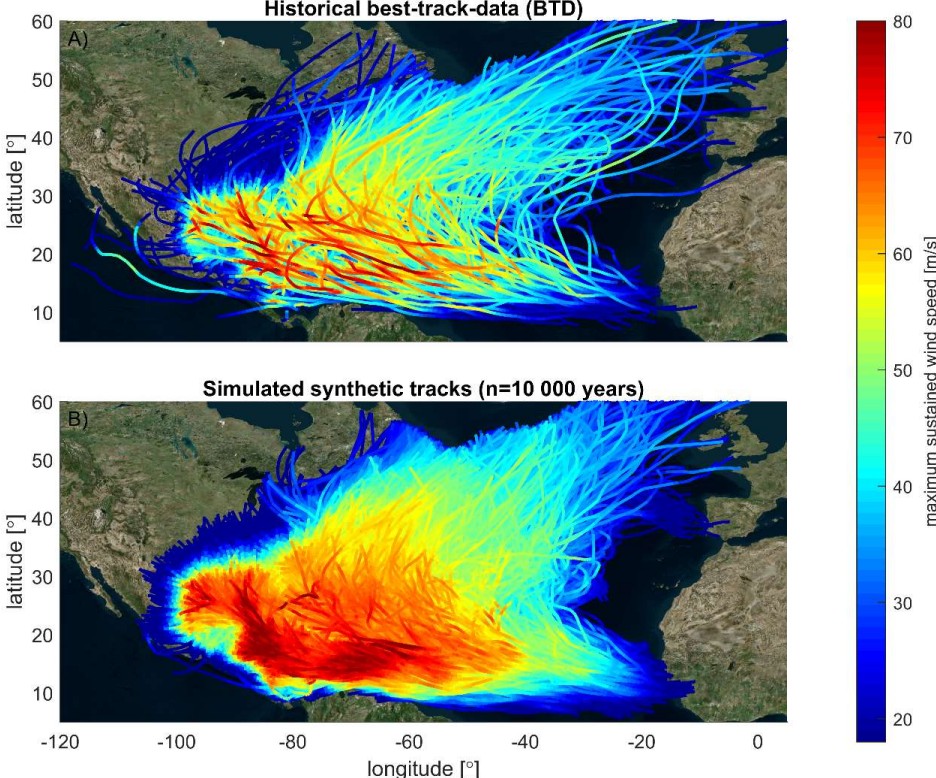

**Figure 4. Panel above (A): overview of historical tropical cyclone tracks in the IBTrACS database for the period 1886-2019. Panel below (B): 10 000 years of simulated tracks with TCWISE. Colors indicate the maximum sustained wind speed of the TC core. Note that the maximum sustained wind speed is the maximum wind speed per TC and not the same as the wind field and/or wind swaths.**
5  **© Microsoft Bing Maps.**

The average yearly occurrence of historical and synthetic TCs is presented in Figure 5. A high occurrence of TCs in the GoM, Caribbean and along the east coast of the US is observed for both historical and simulated tracks. The simulated occurrence is quite similar but, as expected, more smoothened for the synthetic tracks. The Kirchhofer metric score for occurrence confirms the matching of the patterns with a high score of 0.926. This gives confidence that TCWiSE produces synthetic TCs with a
10  similar occurrence rate as historically observed.

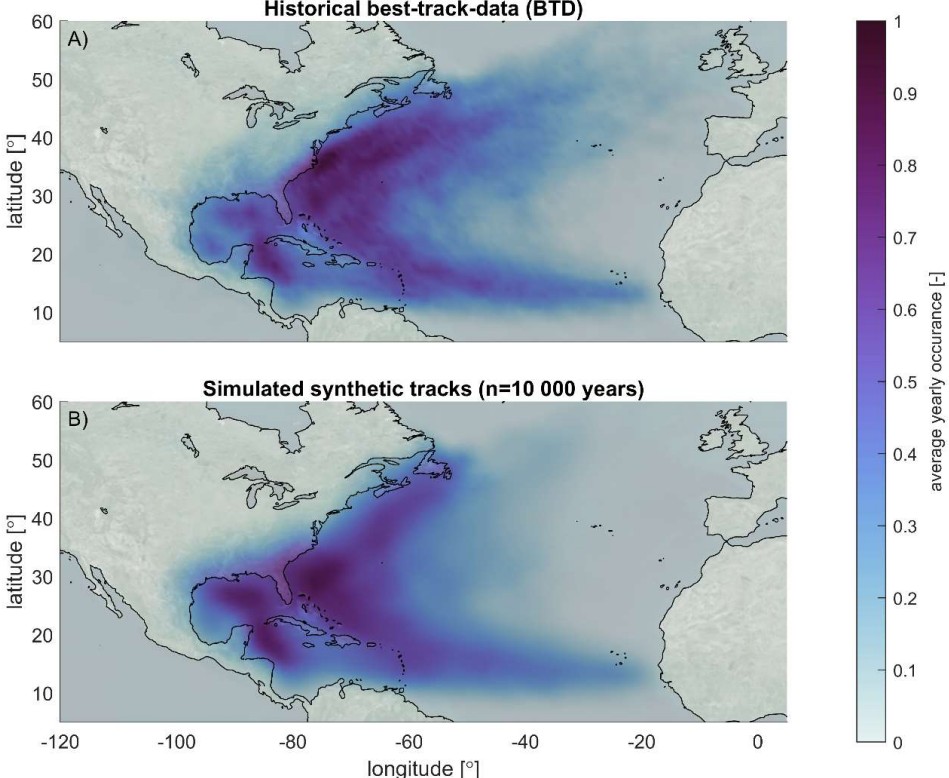

**Figure 5. Average occurrence of historical (BTD) (upper panel; A) and simulated TCs with TCWiSE (bottom panel; B) for the NA basin. Occurrence is based on TCs within 200km per grid cell for historical TCs from 1886-2019 and 10 000 years of simulated events. © Microsoft Bing Maps.**

5    The generation of synthetic TCs includes three distinct parameters which can be compared between the historical and synthetic tracks namely: forward speed (c), heading (θ) and maximum sustained wind speeds ($v_{max}$). The CDFs are presented for these parameters in Figures 6 to 8 for the nine locations as shown in the map in Figure 2. Visually the CDFs of the synthetic data appear to match those of the historical data rather well. nMAE of the forward speed (Figure 6) vary between 0.02 – 0.20 with an average RMSE of around 0.43 m/s and with a bias of +0.31 m/s. For example, Location 3 (WS) and Location 9 (NW) have

10    a larger error due to the positive bias. Statistical errors in the headings (Figure 7) are generally small too. Location 2 and 9 have larger nMAE than the other control locations (possibly due to the effect of land), while Location 7 and 8 have the lowest errors. The nMAE of maximum sustained wind speed (Figure 8) varies between 0.00 – 0.04 with, on average, a RMSE of around 3.62 m/s and with a bias of -3.10 m/s. These error statistics do reveal a general tendency for larger deviations closer to land, but give confidence in the synthetic generation and propagation of the TC.



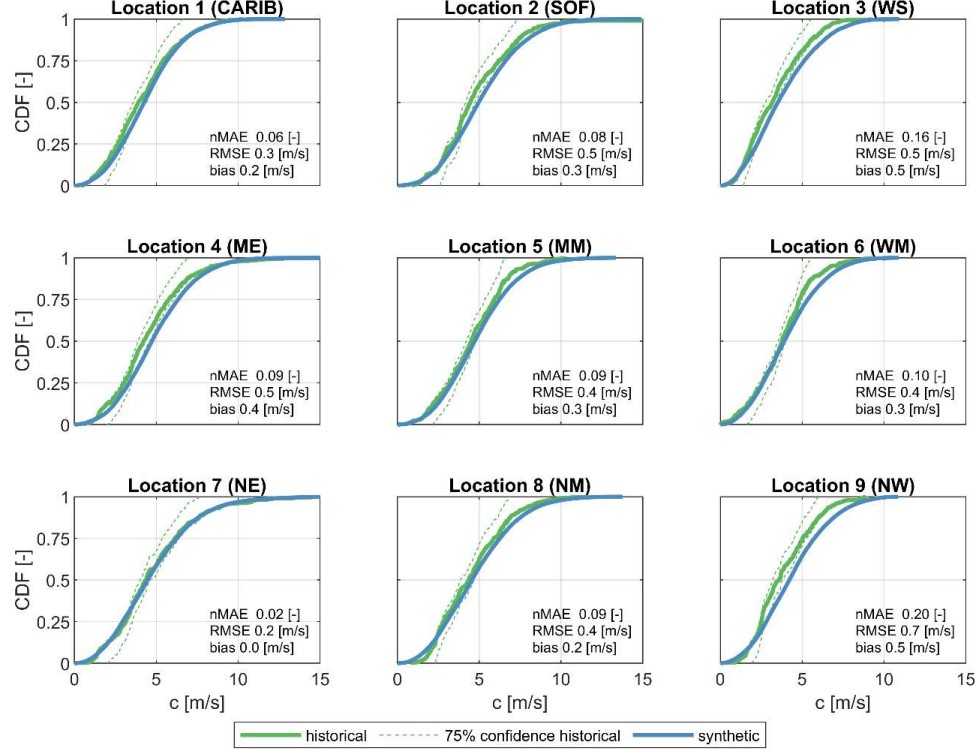

**Figure 6. CDFs of forward speed (c [m/s]) of historical (green line) and synthetic (blue line) TCs at 9 locations within the NA basin, as shown in the map in Figure 2. The 75% confidence interval (dashed green line) of historical data is also shown. Historic data are based on data available between 1886 and 2019, while synthetic data are derived from 10 000 years of simulated events with TCWiSE.**

5      **Data points within 200 km from the control location are included in the analysis both for the historical and synthetic data.**

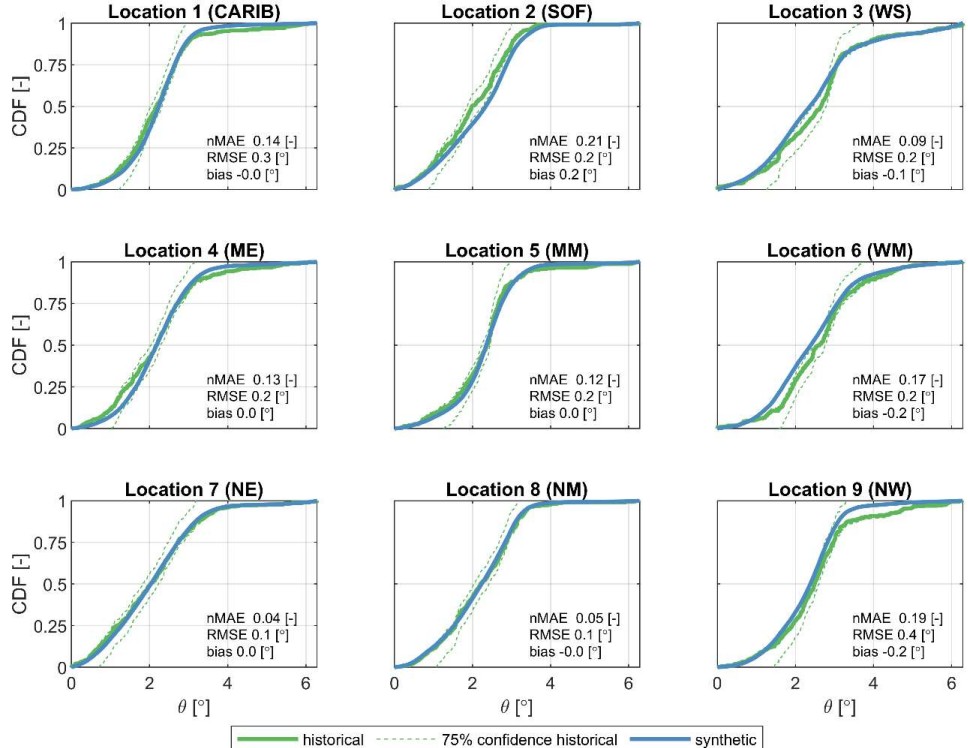

**Figure 7. CDFs of heading (θ [°]) of historical (green line) and synthetic (blue line) TCs at 9 locations within the NA basin, as shown in the map in Figure 2. The 75% confidence interval (dashed green line) of historical data is also shown. Historic data are based on data available between 1886 and 2019, while synthetic data are derived from 10 000 years of simulated events with TCWiSE. Data points within 200 km from the control location are included in the analysis both for the historical and synthetic data.**



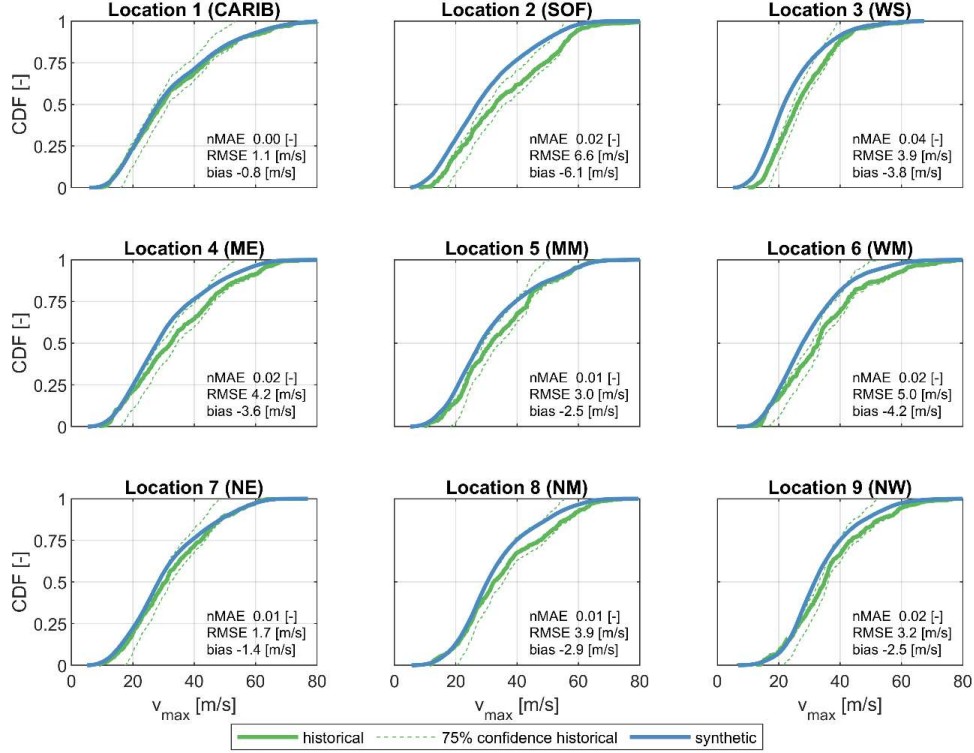

**Figure 8. CDFs of maximum sustained wind speed ($v_{max}$ [m/s]) of historical (green line) and synthetic (blue line) TCs at 9 locations within the NA basin, as shown in the map in Figure 2. The 75% confidence interval (dashed green line) of historical data is also shown. Historic data are based on data available between 1886 and 2019, while synthetic data are derived from 10 000 years of**
5 **simulated events with TCWiSE. Data points within 200 km from the control location are included in the analysis both for the historical and synthetic data.**





*Track termination*

Historical and simulated termination probability is shown in Figure 9. In TCWiSE, cyclone termination is defined as the last point of TC that is obtained from the BTD. The figure shows that historically there is a large probability of termination at the East coast of Canada (i.e. Nova Scotia and Island of Newfoundland) (see e.g. Elsner et al, 1999) and the East coast of Mexico.

5    At some cases, TCs terminate after landfall in the US or while propagating on the Atlantic Ocean. Visually, the historical and simulated termination does not match very well. The reasons for deviations are because termination can be triggered by several different physical processes and is thus not so closely related to the input data. In particular, in TCWiSE, synthetic TCs can get terminated due to a low temperature of the ocean and/or low wind speed on land. Hence, the differences in this comparison can be explained due to the schematization of the physical processes which lead to a different TC termination in TCWISE than

10   based on the historical probability alone. Moreover, errors from the previous steps in the TC life (i.e. genesis location, propagation, etc.) will be compounded in the track termination. The comparison between historical and simulated termination probability was quantified by using the Kirchhofer metric score for termination, which provided a value of 0.622 (compared to 0.967 for genesis and 0.926 for occurrence).

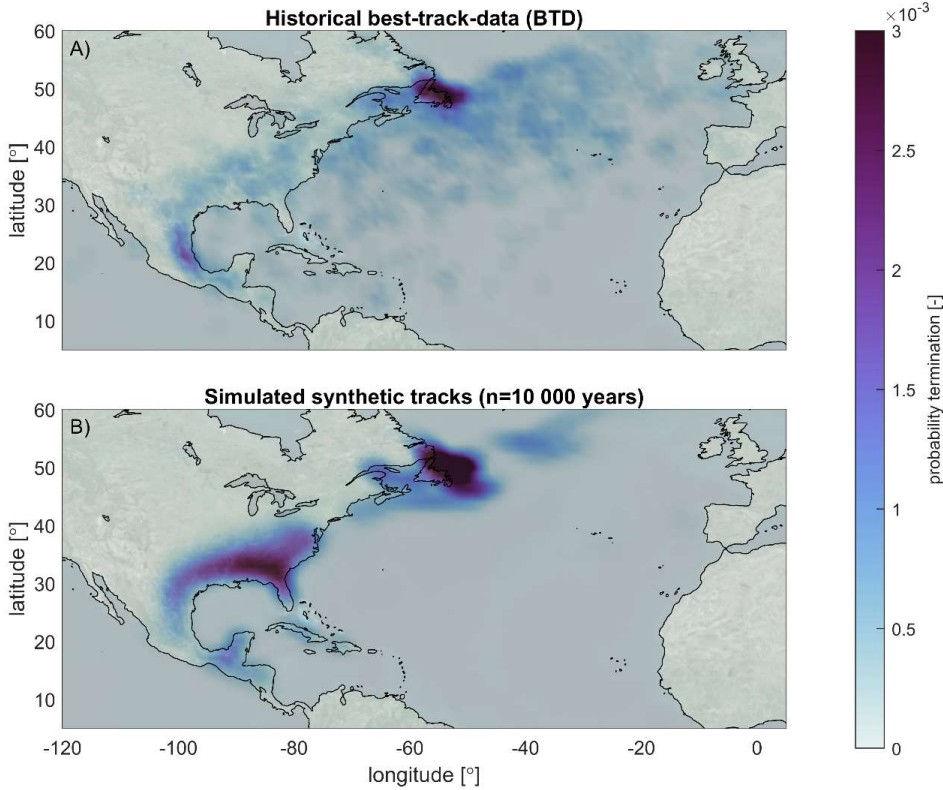

**Figure 9. Termination probability of historical (BTD) (upper panel, A) and simulated TCs with TCWiSE (bottom panel, B) for the NA basin. Occurrence is based on TCs within 200 km per grid cell for historical TCs from 1886-2019 and 10 000 years of simulated events. © Microsoft Bing Maps.**

5    **3.3 Computed and historic TC maximum wind speeds**

*Observed extreme wind speeds*

Figure 10 presents the non-parametric 10-year return value estimates TC wind speed for the GoM based on synthetic TCs. Cooler colors depict lower TC wind speeds and warmer colors higher wind speeds. The circles indicate the non-parametric estimates based on buoy observations for the same return period; given that the observations cover about 40 years they are the

10    fourth-highest ever recorded value. The figure shows how the general patterns of higher wind speeds in the central GoM and lower values near land, as shown by the data, are reproduced correctly by TCWiSE. The model computed values are biased high (i.e. overestimation) for stations near land. This is most likely due to land-related processes not being fully accounted for in TCWiSE. Also, the data scarcity (sub-sampling) affects the estimates from the observations.

Figure 11 presents a comparison between observed and TCWiSE computed TC extreme wind speeds, for different return periods, at nine locations throughout the GoM, both based on historical and synthetic tracks. As could already be seen in Figure 10, there is some scatter between observed, historical and synthetic TC wind speeds. For example, the peak in the observed wind speed, in particular of larger return periods, at East GoM (Figure 11A) and Middle GoM (Figure 11B) are

underestimated by both the historical and synthetic TCs. These are respectively peaks corresponding to Hurricane Rita (2005) and Hurricane Kate (1985). Based on the observation record of 40 years, the non-parametric return period estimate is of 40 years, whereas TCWiSE indicates that the return period associated with those events is higher. The cause of the large difference between observed wind speeds and values derived from historical and synthetic TCs wind speed for West GoM (Figure 11C) is unclear. On the other hand, at Venice, FL (Figure 11E) and Port Arkansas, TX( Figure 11I) seem to be overestimated by the

historical TCs and synthetic TCs which could be related to unresolved land-related processes.

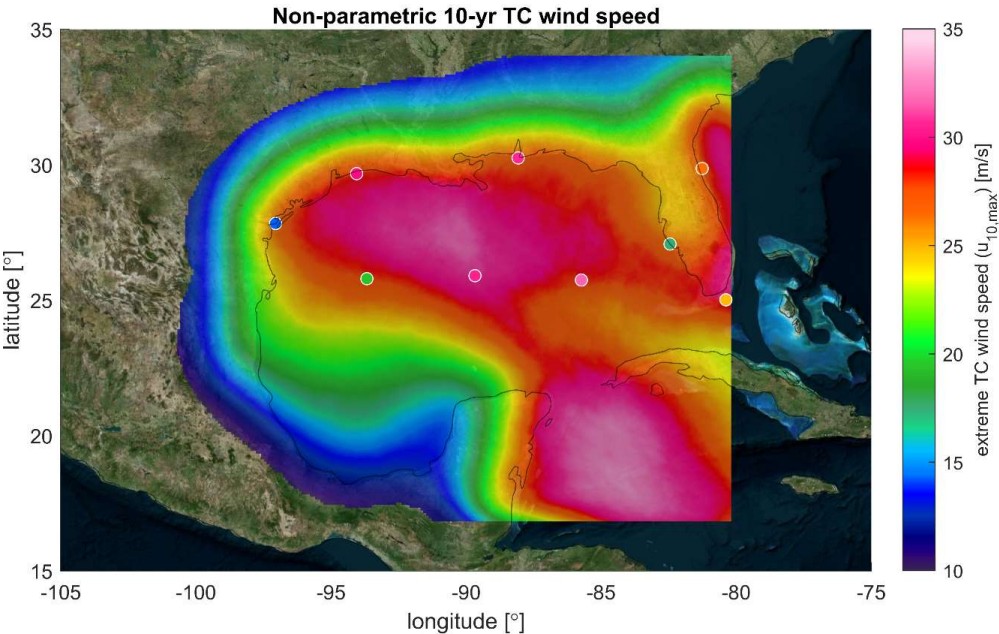

**Figure 10. Model estimates for non-parametric empirical estimate of 10-year TC wind speed return values based on extreme wind speeds based on 10 000 years of TCWiSE computations. The white circles indicate observed TC wind speed extremes based on NDBC**
**wave buoys/NOAA data. All wind speeds are in m/s, 10 minute-averaged and determined at a 10-meter height. © Microsoft Bing Maps.**

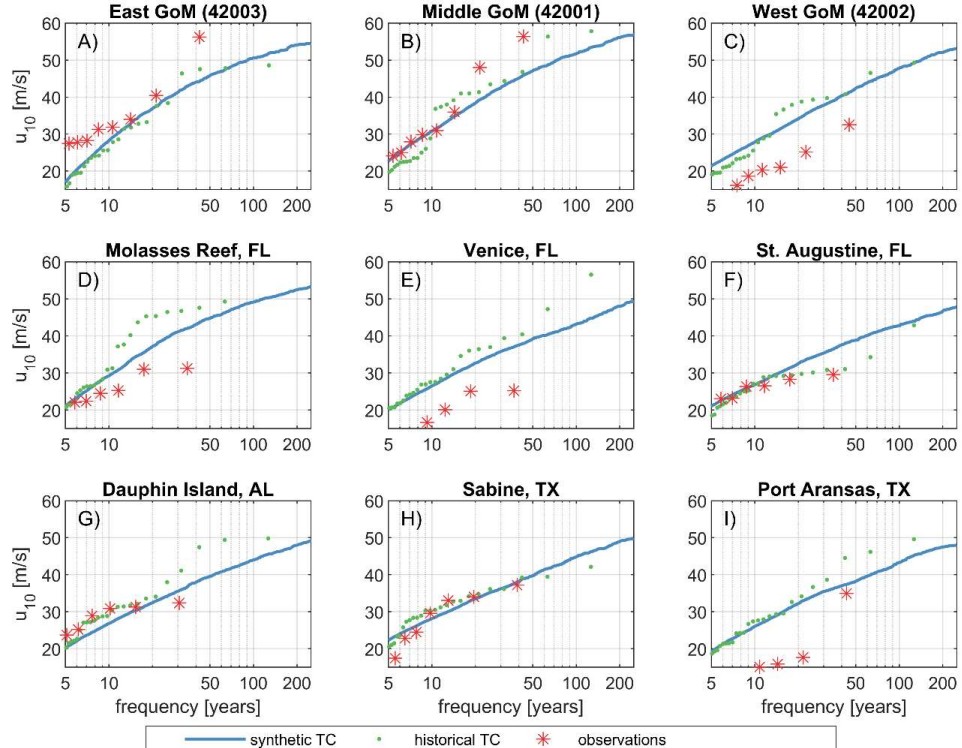

**Figure 11. Observed and TCWiSE computed TC extreme wind speeds for different return periods. Red stars are observed events from NDBC and NOAA wave buoys. Green dots are historical TCs and blue line are synthetic modeled events. All wind speeds are in m/s, 10 minute-averaged, on 10-meter height).**

*Modeled extreme wind speeds*

Figure 12 presents the 1 000-year parametric TC wind speed for the GoM, estimated by fitting a GPD to the POT of the generated data. The figure shows a spatial pattern similar to that of the 10-year non-parametric TC wind speeds (Figure 10). The highest values are found in the Caribbean Sea and central GoM. Lower values can be found in the Florida Panhandle / Northwest Florida and in the southwest of the GoM. This is in line with literature (e.g. Neumann, 1991). Computed occurrence

rates are also in line with NOAA values for both hurricanes (>64 knots) and major hurricane (>96 knots) within 50 nautical miles (92.6 kilometers). Occurrence rates for major hurricanes (>96 knots) are the highest for South Florida and Louisiana with a respective return period of 17-20 years. TCWiSE estimates are 19-22 years.

Figure 13 presents the estimated TC wind speed return value swaths versus coastal milepost which starts at Cancun, Mexico and goes across the GoM in a clockwise orientation. Several return periods are depicted in different colors. Moreover,

TC wind speed is presented both in 10-minute averaged in m/s and 1-minute in knots. Saffir–Simpson hurricane wind scale

Natural Hazards
and Earth System
(SSHWS) is included as well. TCWiSE simulation indicates for a return period of 10-year TC wind speed of around 30 m/s (close to SSHWS-1) near Cancun and large stretches of the US coastline. For a return period of 1 000-year, this increases to values around 60 m/s (around SSHSS-4). Generally, values near Villahermosa are the lowest for all GoM. Vickery et al. (2009) reported maximum gust TC wind speeds with a return period of 100-year that vary between 33-57 m/s (using a conversion

5    factor of 1.23 based on Harper et al., 2010). TCWiSE indicates values of the same order of magnitude but with less spatial-variability.

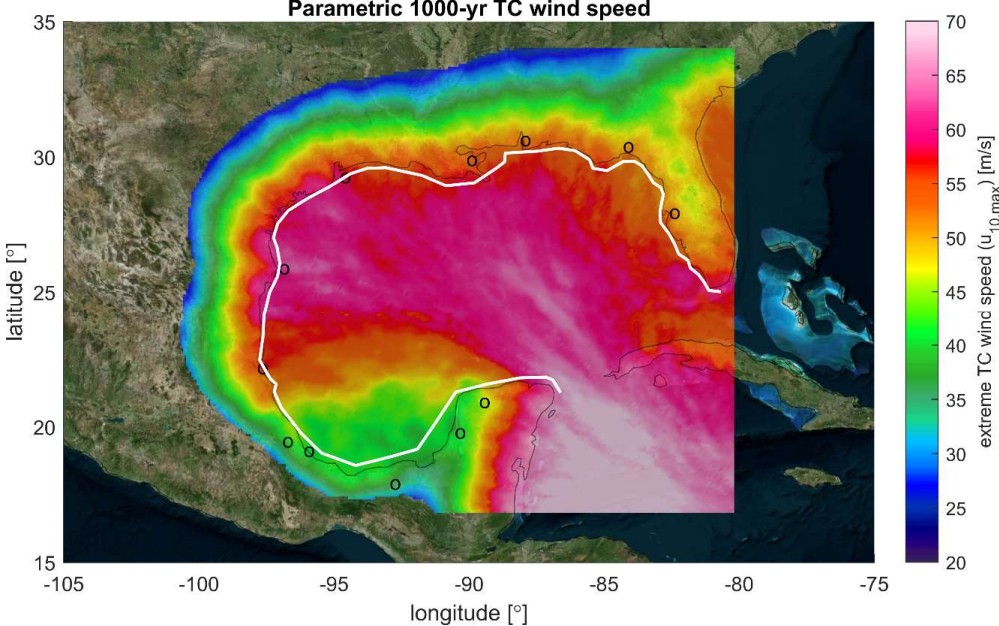

**Figure 12. Model estimates for the parametric empirical estimate of 1000-year TC wind speed return values based on extreme wind**
10   **speeds based on 10 000 years of TCWiSE computations. All wind speeds are in m/s, 10 minute-averaged and on 10-meter height.**
**Black dots are the location of cities as plotted in Figure 10 and 13. © Microsoft Bing Maps.**

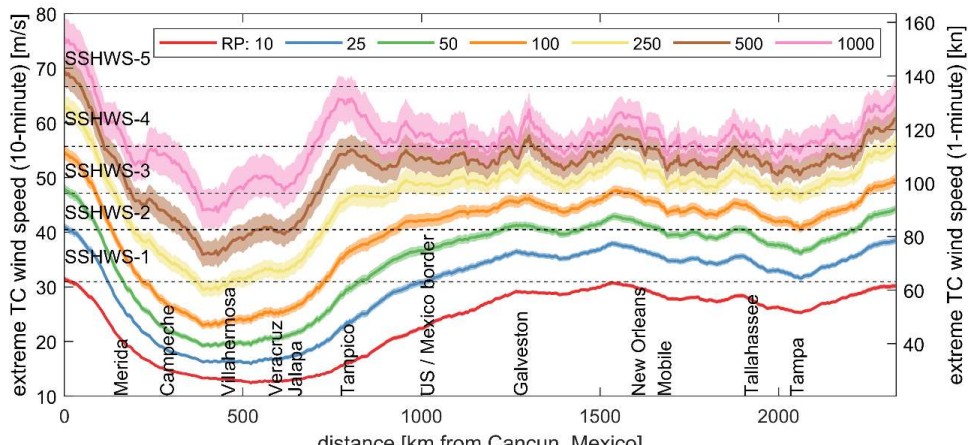

**Figure 13. TCWiSE 10-, 25-, 50-, 100-, 250-, 500- and 1 000-year return value estimates of wind speed. All wind speeds are in m/s, 10 minute-averaged and on 10-meter height (left axis) or 1-minute averaged in knots (right axis). Cities on the x-axis are also depicted in Figure 12 as black circles. Milepost is presented in the same figure as a white line. Shading shows the 5/95% confidence interval.**
**SSHWS-value indicates the corresponding Saffir-Simpson hurricane wind scale.**

## 4. Discussion

For clarity, discussion points have been grouped under two main topics: the TCWiSE tool and the validation study.

### 4.1 The TCWiSE tool

The philosophy which guided the development of TCWISE is to release an open-source tool, giving modelers full control over
the track generation, propagation and termination. However, this makes TCWiSE also more sensitive to input errors compared to pre-generated global synthetic TC data products (e.g Bloemendaal et al., 2020). However, the strength of this approach is twofold. First of all, this allows the user of TCWiSE to rigorously calibrate and validate assumptions within the code for its own case study site. Secondly, due to the flexible Matlab coding language, it allows easy adjustments of the tool and implementation of additional processes. For example, stochastic rainfall was recently added to the original code by Bader
(2019).

TCWiSE is an almost completely data-driven tool to simulate synthetic TCs. As such, output values are highly dependent on the (historical) input data and not the physical processes describing the genesis, propagation and termination of these TCs. If cyclone characteristics are expected to behave identically as over the last decades, this method has been proven accurate for the determination of extremes. However, climate change is expected to influence future TCs frequency and
intensity (e.g. Knutson et al. 2010). This can already be accounted for by a heuristic factor to adjust both the frequency and




intensity of the TC (or other variations implemented by the user) to reflect changes due to climate change. Other methods, e.g. by either adjusting the KDE or by using datasets derived by global climate models, are currently being investigated.

The effect of land on intensity can be taken into account either directly via the conditional-dependent KDE or landward decay based on De Maria and Kaplan (2005). The latter is beneficial since TC information on land contaminates the
KDE of intensity. In particular, due to the applied search range methodology, information from decreasing winds on land start to affect winds on the water. The downside of this method is that this does introduce an additional calibration coefficient for the user and larger deviations in the termination. Moreover, TCWiSE does not include a boundary layer model which means that the physical wind response to variable surface drag and terrain height is not included. Done et al. (2020) has however shown that the output of parametric wind models can be used to simulate the near-surface spatial wind fields of landfalling
TCs, accounting for terrain effects such as coastal hills and abrupt changes in surface roughness due to coastlines and forested or urban areas.

In TCWiSE, track termination can be either be purely based on historical track termination or via additional formulations based on user-definable cut-off wind speed and/or SST. While these additional formulations were of importance to get the track evolution (and thus associated coastal hazards) simulated correctly, they do result in deviations of simulated
track termination compared to historical data. However, arguably, track termination is not of importance for the simulation of coastal hazards and therefore this is deemed an acceptable trade-off for the more reproductive skill in the track evolution.

TCWiSE does not take into account errors in the wind fields and its associated impact on the confidence interval for the computed return periods for wind speeds. Nederhoff et al. (2019) showed that the Holland wind profile in combination with reliable estimates of the TC geometry (i.e. the radius of maximum wind and gale force winds) to calibrate the wind profile
wind, has a median root-mean-square-difference of 2.9 m/s. Other approaches (e.g. Vickery et al., 2009) do include error estimates in their estimates of the extreme winds and conclude that uncertainty in the estimated 100-year return period wind speed varies from about 6% along the Gulf of Mexico coastline, which corresponds to about +/- 3-5 m/s.

### 4.2 Validation study

Validation results across the NA basin and in particular the GoM have shown that TCWiSE can reproduce the main patterns
seen in the BTD, wind observations and literature. It does seem however that synthetic TC tracks have a less clear southwest-northeast orientation in heading on the North Atlantic Ocean. This could be related to the lack of physical description of the Jetstream given that TCWiSE is a purely data-driven tool and does not include specific processes to steer TC propagation.

A comparison of similarities in spatial patterns between synthetic tracks and historic tracks, evaluated by means of the Kirchhofer metric score, show that TCWISE is able to correctly reproduce genesis and TC occurrence while differences
were found for the TC termination. These differences can be attributed to the fact that TC termination can get triggered by several processes in TCWISE and is not just related to the historical probability of termination. At the same time, this has a relatively minor effect on the track evolution and consequently coastal hazards.



The comparison of CDFs of forward speed, heading and maximum sustained wind speed of historical and synthetic tracks show in general a good agreement for the different stations. While differences between observed and modeled CDFs are apparent, results of the goodness-of-fit tests are generally acceptable (Figure 6-8) with mean nMAE of 0.08. More classical statistical tests such as Kolmogorov–Smirnov were not presented here and often reject the null hypothesis that the observed

and modeled data are from the same distribution. This is related to the methodology of providing inputs to the Markov chains. While this method resulted in reliable probability distributions, it also smoothed out some local spatial patterns and therefore resulted in differences at the nine control locations. Arguably, locally patterns in the BTD (features < 500 km) could well be subject to a sampling error and not necessarily a feature of the TC climate we aim to reproduce.

All BTD, since 1866, has been included as a basis for the generation of the synthetic tracks. Especially for pre-satellite

records, errors in the BTD can be quite significant so previous studies (e.g. Holland, 2008) selected a specific subset of the BTD to ensure the quality of the data and remove potential inconsistencies. However, the advantage of including all data entries is that the derived TC climate is more widely defined (i.e. larger parameter space). It is however easily possible in TCWiSE to only include tracks from more recent years.

## 5. Conclusions

A new methodology and highly flexible open-source tool has been developed with which synthetic tropical cyclones (TC) can be generated and used for subsequent analysis of (coastal) hazards. In particular, TCWiSE handles track initialization, evolution, and termination based on historical TC information. Subsequently, the tool creates a spatially-varying wind field based on the Holland wind profile calibrated for TC geometry. Lastly, TCWiSE computes non-parametric and parametric wind swaths for user-definable return periods.

The validation study for the North Atlantic and in particular the Gulf of Mexico showed reliable skill in terms of track initialization and evolution compared to the historical BTD. A more detailed assessment of the goodness-of-fits at nine control location showed that normalized errors are generally smaller than 10%. Extreme wind speeds show agreement for more frequent return period, with possible deviation for the most extreme cases. This is the result of biases associated with the scarcity of observed data.

TCWiSE can be useful in a variety of applications.  Improved estimates of extreme TC conditions, can lead to a better quantification of coastal hazards (e.g. extreme storm surge levels and waves), and consequent risks and damages resulting from these hazards. Similarly, an improved assessment of those hazards can help the design of appropriate adaptation measures. Other fields of application may vary from improved design parameters for offshore structures to navigation. In all these types of applications, the flexibility of TCWiSE to tailor the synthetic TC generation to user-specific needs and questions makes the

tool very suited for coastal engineers. The application of the tool for determining coastal hazards will be presented as part of a separate paper currently under preparation (Leijnse et al., in prep).





*Code and data availability.*

TCWiSE is freely available for other researchers and consultants. The repository consists of the Matlab code and required input data (such as BTD and SST) map. Registration is required before getting access to the subversion (URL will be made available after acceptance of the paper).

*Author contributions.*

KN, TL, JH and MvO developed the Matlab code for computing synthetic tropical cyclone tracks. SC and AG supported the development of the idea and the writing of the paper.

10   *Competing interests.*

The authors declare that they have no conflict of interest.

*Acknowledgements and financial support*

The authors thank the Deltares research programs 'Flood Risk Strategies', 'Planning for disaster risk reduction and resilience'
15   and 'Extreme Weather Events' which has provided financings to develop TCWiSE and write this paper. Also, may thanks to Deepak Vatvani for all his inspiring cyclone work over last decades at Deltares and Bjorn Robke for Figure 2.



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
