# Peer review of "Simulating Synthetic Tropical Cyclone Tracks for Statistically Reliable Wind and Pressure Estimations"

_Natural Hazards and Earth System Sciences, 2020_

## Referee Comment (RC1) · Anonymous Referee #1 · 20 Sep 2020

This article describes a new open-source tool (TCWiSE) for the simulation of synthetic TC tracks. The utility of this kind of approach is that in many areas the frequency of the TC is too low to provide adequate historical statistics. Though similar works exist, an important advantage of TCWiSE is that it is open-source, which potentially makes it a useful tool for research groups active in risk analysis.

The paper is clear and well written, though some details of the algorithm require additional explanation (see below).

An important limitation that I see in the applicability of this approach for studies on climate changes, is that it does not consider explicitly variables like SST in the cyclo-

[Figure]

genesis. Hopefully, this limitation will be overcome in future releases.

Overall, I recommend the paper for publication after a moderate revision. Below, some more detailed comments.

p2, l8: how reliable are very old data? Can we assume that the frequency of TC 100 years ago was similar to that of today?

p2, l18: the term "heading" should be defined

p4, l8: it could be useful to add, in the future, 1st order estimations of the ocean variables as well.

p4, l16: "number of points needed per KDE" does not read well. You mean, the kernel size?

p4, l17: "The user can also define bulk ... climate changes." But to do so I should assume a dependency between TC frequency and climate variables such as SST, or build a further statistical model to infer it. I believe this would be better done inside TCWiSE, please consider it as a future development.

p4, l25: .. poisson distribution ... this is not very clear. how do you define the poisson dist? monthy or seasonally?"

p5, l6: .. sea surface temperature (SST) .. I guess SST is somehow estimated by TCWiSE? How?

p5, l12: "last track". How does the algorithm decides that it has to generate nothing else? At the end of its time horizon?

p5, l16: "create wind swaths", in fig 1 it is said that is done by means of POT GPD. would you clarify how?

p5, l16: The difference between wind swaths and maximum wind is not very clear.

p7, l4: "temporal variability of genesis locations or other input parameters are included

in the tool" but you mentioned earlier that a Poisson dist is used to model the seasonal dependency (how?)

p7, l9: "Genesis location in ocean surface temperatures less than a user-definable value .." this sentence is not well formulated

p7, l23: "The KDE that is sampled are constructed for each grid point based on input data within a specific search range." this sencence does not read well

p8, l25: "... not completely similar to the historical ..." maybe this could also depend on the way the termination in historical data is defined? Do all the agencies define the TC termination in the same way?

p8, l30 & p9, l3: see my previous comment for p4, l17

p9, l11: wind swaths: it is still a bit unclear what the wind swaths are and how you do generate them - on what variable is the GPD fitted?

p11, l30: "MAE": you mean, the MAE between historical and TCWiSE cdf? Please clarify

p11, paragraph 3.3.1: I would suggest adding formulas with the error indicators used

p12, l15: "the genesis patterns ..." this sentence does not read well

figure 4: "is the maximum wind speed per TC and not the same as the wind field and/or wind swaths" again, the difference should be explained

p15, l9: "for example" looks out of context

p15, l9: the TCWiSE bias of c vs historical looks generally slightly negative. Is it only in this case?

figure 7: the scale looks in radiants rather than in deg

figure 8: the bias in TCWiSE max wind looks slightly positive. Is it a systematic tendency or is it random?

p19, track termination. To what extend may these differences depend on the uncertainty of historical data on track termination?

p20, l7: estimates *of* TC winds

p21, l2: How many TC were used for the estimation of the extremes on the historical? How do you ensure the extremes on the historical are compatible with the ones on the synthetic tracks?

p21, l5-9: the authors should mention here that the large differences are due to the differences between the historical, used to fit the model, and the observations. They should also mention, earlier, that the historical data are model data, and not observations

p24, l10: "this makes TCWiSE also more sensitive to input errors compared ...", unclear why this should be: the algorithm used by other authors may as well be sensitive to input errors

p25, l2: ".. using datasets derived by global climate models .." you mean, CMIPX? How? These models are quite unable to represent properly the TCs. That's a reason why statistical tools like this can come in handy

p25, l20-22: this sentence is a bit unclear and full of repetitions

p25, l25: " It does seem however that synthetic TC tracks have a less clear southwest-" from what do you see this?

p25, l27: subtitute Jetstream with "climate dynamics"

p25, l30: "These differences can be attributed to the fact that TC termination can get triggered by ..." this sentence does not read well

If the tool is open source, I believe it would be useful to provide a link to a code repository

---

## Referee Comment (RC2) · James Done (Referee) · 30 Sep 2020

General Comments:

This study describes a new tool for the creation of synthetic tropical cyclone (TC) tracks based on Markov-chains. While the overall approach doesn't differ substantially from previous studies, its novelty lies in its flexibility. The approach is agnostic to the choice of input data and can therefore be applied to any global TC basin. It can also ingest historical track data or even track from global climate models. The tool permits many options for run-time configuration and is designed to be flexible to allow a variety of applications from scientific research to coastal engineering applications. Perhaps the

most unique aspect of its functionality is the inclusion of climate change parameter choices such as options for TC intensity and frequency shifts.

The paper includes an in-depth tool demonstration and evaluation for the North Atlantic basin with focus on the data-rich Gulf of Mexico. This includes a suite of robust statistical significance tests and a valuable combination of point-location and spatial evaluations. Overall the model performs well.

I fully expect that this open source tool will become widely used both as a research tool and a risk assessment tool. I congratulate the authors in making it available. The paper is generally well written. But there are a few grammatical quirks and awkward word choices that can be corrected by a thorough review of English grammar. The subject matter is appropriate for NHESS and is worth being published after my comments below have been addressed.

Specific Main Comments

1) I agree that synthetic track simulation adds events and overcomes the sampling problem. But these need to be interpreted in the correct context. These synthetic tracks are constrained to reproduce the statistics of the historical record. This means that this tool would not, for example, produce a Hurricane Sandy-like track before Sandy occurred in the historical record. A physical model on the other hand has the potential to produce physically credible but not observed track behaviors. I suggest making this point in the discussion.

2) Another limitation of the tool is the assumption of stationarity in the historical record. We know that change has been detected in some TC characteristics in some regions (Knutson et al. 2019). Perhaps this limitation can be stated in the discussion.

Knutson, T., Camargo, S.J., Chan, J.C., Emanuel, K., Ho, C.H., Kossin, J., Mohapatra, M., Satoh, M., Sugi, M., Walsh, K. and Wu, L., 2019. Tropical cyclones and climate change assessment: Part I: Detection and attribution. Bulletin of the American Meteorological Society, 100(10), pp.1987-2007.

3) I'm glad to see the option to include inland wind decay of Kaplan and De Maria (1995) in addition to the implicit decay through the KDE of Vmax. But it's important to state in the manuscript that at-sea winds will still extend inland before the TC center crosses the coast and the Kaplan and De Maria wind decay turns on. I think this is a possible reason for your high bias in 10-year return winds in some coastal regions (Fig. 10).

4) Section 2.5: Can you explain in more detail how asymmetry is considered? The Vmax in BTD is ground-relative and so includes a component of asymmetry. Did you remove the component of asymmetry from the BTD Vmax before creating the synthetic tracks and running the symmetric Holland model (and then add asymmetry back to the spatial wind field afterwards)?

5) There are a couple of notable omissions from the reference list. Arthur (in review) has a paper under discussion at NHESS that describes a synthetic track model that has similar functionality to this study. Lee et al. (2018) published a synthetic track model that differs from your data-driven approach by accounting for environmental drivers of TC behavior.

Arthur, W. C.: A statistical-parametric model of tropical cyclones for hazard assessment, Nat. Hazards Earth Syst. Sci. Discuss., https://doi.org/10.5194/nhess-2019-192, in review, 2019.

Lee, C.Y., Tippett, M.K., Sobel, A.H. and Camargo, S.J., 2018. An environmentally forced tropical cyclone hazard model. Journal of Advances in Modeling Earth Systems, 10(1), pp.223-241.

Specific Minor Comments

1) Abstract, lines 10-12: This sentence makes the point that short historical records may not represent the parent population. This is a valid point but I suggest not using

the term 'future TCs' in this sentence because that implies climate change and non-stationarity which is a separate issue.

2) Introduction: The sentence spanning lines 27-29 about first and second order effects doesn't appear to fit well in this paragraph about extreme value modeling.

3) The introduction talks a lot about the multi-hazard nature of TCs but then the paper describes a tool for TC wind only. I suggest toning done the discussion of surge, waves and rainfall in the discussion and just mentioning it briefly.

4) To improve the flow of the introduction, can the point about the need for a larger sample size be made just once? It is currently discussed twice in the first and third paragraphs.

5) Section 2.2. Why not choose a threshold of 17ms to include all Tropical Storms rather than is seemingly arbitrary 25 m/s?

6) Section 2.4.3. What are the units '10kn/s'. Do you mean knots?

7) Can you comment on the computational performance of the tool? How long does it take to run 10,000 years of the North Atlantic, for example?

8) Fig. 5. Would it be useful to additionally plot the difference field to highlight the differences discussed in the main text?

9) In Section 3.3, use of the fourth-highest recorded value for the 10-year return wind will probably be noisy. Would it be better to fit an extreme value distribution to the observations to estimate the return value? This may produce a better agreement with the model.

10) Figure 11: Can you clarify what the historical TC wind data are please? Is it Holland model run along historical tracks?

11) The description of Fig. 11 in main text has 'Port Arkansas'. The correct name is Port Aransas. 12) Figures 10 and 12: Please state the grid spacing used in these

figures.

13) Section 4.2. I don't see what you are referring to about the synthetic TC tracks having a less clear southwest to northeast orientation. I think this needs to be quantified in some way or excluded from the manuscript.

14) The Hoek (2017) reference was incomplete in my pdf version.

15) The Bader (2019) reference is missing from the reference list.

---

## Author Comment (AC1) · 28 Oct 2020

Rebuttal letter manuscript "Simulating Synthetic Tropical Cyclone Tracks for Statistically Reliable Wind and Pressure Estimations"

Dear editor, dear reviewers,

On the July 31, 2020, we have submitted the following manuscript to the Journal of Natural Hazards and Earth System Sciences titled: "Simulating Synthetic Tropical Cyclone Tracks for Statistically Reliable Wind and Pressure Estimations" (MS No.: nhess-2020-250). On the October 7, 2020, we were informed that the open discussion was

completed. In total, we received comments by two reviewers which provided a very positive feedback on the work done and valid suggestions. We would like to acknowledge their time and efforts, which have led to an improvement in the quality of our manuscript. Below you find a point-by-point reply to all specific questions and suggestions. Attached you also find the revised manuscript with the changes made to address the review comments tracked.

Kind regards, Kees Nederhoff — Anonymous Referee #1

General Comments:

1. An important limitation that I see in the applicability of this approach for studies on climate changes, is that it does not consider explicitly variables like SST in the cyclogenesis. Hopefully, this limitation will be overcome in future releases

We agree with the reviewer that climate change studies are of vital importance to our research field. TCWiSE does indeed not consider SST in the cyclogenesis but it can be used to study the effects of climate change using data from for instance IPCC studies on changes in the intensity and/or frequency distributions (Page 9 Lines 5-10; later in this rebuttal will be referred to as P9 L5-10).

Specific comments

1. p2, l8: how reliable are very old data? Can we assume that the frequency of TC 100 years ago was similar to that of today? The tool at the moment assumes stationarity, with historical data being assumed to describe the current climate. As stated in the reply to the general comment above, it possible to take into account a heuristic implementation of a factor on both the frequency and intensity. In order to address the first question we have added a statement on the increasing quality of historical data to the introduction (P2 L10-13).

2. p2, l18: the term "heading" should be defined We have added the definition of heading (P2 L22).

3. p4, l8: it could be useful to add, in the future, 1st order estimations of the ocean variables as well. TCWiSE only computes TC tracks and winds. We have at the moment no immediate plan to include, within TCWiSE, the effects of these winds on other variables, such as water level and surface currents. TCWiSE does however support the creation of output file in a format that can be used directly in open source models (currently only Delft3D4 and Delft3D-FM are supported including flow and wave). In addition, TCWiSE does take the ocean variable SST into account to determine track termination.

4. p4, l16: "number of points needed per KDE" does not read well. You mean, the kernel size? Adjusted as suggested (P4 L22-23).

5. p4, l17: "The user can also define bulk ... climate changes." But to do so I should assume a dependency between TC frequency and climate variables such as SST, or build a further statistical model to infer it. I believe this would be better done inside TCWiSE, please consider it as a future development. TCWiSE is a purely data-driven approach, with no ability to simulate the TC generation physical processes. This means that this information needs to be input from other sources. At the same time, this can also be seen as a flexible aspect of the tool, since no assumptions are being made on the TC generation process.

6. p4, l25: .. poisson distribution ... this is not very clear. how do you define the poisson dist? monthy or seasonally?" Annually and monthly. The Poisson distribution is a discrete probability distribution that expresses the probability of a given number of events occurring in a fixed interval of time or space if these events occur with a known constant mean rate. In TCWiSE, a Poisson distribution is being used for the number of events per year, with the distribution of events during a year being estimated based on a KDE of historical data (see also P4 L30-32 and P5 L1-4 and last paragraph of Section 2.3).

7. p5, l6: .. sea surface temperature (SST) .. I guess SST is somehow estimated by

[Figure]

TCWiSE? How? SST is an input variable. In the presented application, the SST data are extracted from the 1-degree resolution, worldwide monthly average SST map from the International Research Institute of Columbia University (2017) (see also P7 L14).

8. p5, l12: "last track". How does the algorithm decides that it has to generate nothing else? At the end of its time horizon The number of tracks to be generated is also an input variable, more precisely it is determined by multiplying the average number of tracks per year (based on data) with the number of years we want to generate tracks for. We have added additional information to the MS (Section 2.2 item 8).

9. p5, l16: "create wind swaths", in fig 1 it is said that is done by means of POT GPD. would you clarify how? Wind swaths are created based on either non-parametric and/or parametric estimates of the spatially-varying (extreme) wind fields. Non-parametric estimates are determined using the empirical distribution of the collected historical peak (POT) data, the parametric estimates are determined by fitting a GPD distribution to the historical peak (POT) data (see also P5 L24-29).

10. p5, l16: The difference between wind swaths and maximum wind is not very clear. Wind swaths are spatial maps of the maximum (computed) wind speeds per TCs. Hence, they are the same thing. The difference with spatially-varying wind field maps is that these maps have a timestamp. For example, for a 7-day long synthetic TC, we will have 7x8=168 (assuming hourly data) wind speed maps. If we take the maximum of all those maps, we will get the wind swath or maximum wind speed map of that TC. If we do a similar approach to all the synthetic TCs we can start associating probabilities to each wind swath since we have saved this information per grid cell. A more sophisticated approach would be to fit a GPD to the data (i.e. maximum wind speed per TC per grid cell). See also Section 2.2 item 8 where we explain this in the MS.

11. p7, l4: "temporal variability of genesis locations or other input parameters are included in the tool" but you mentioned earlier that a Poisson dist is used to model the

seasonal dependency (how?) We understand the question and agree that the reasoning was not clear in the previous version of the MS. We have changed our explanation in the current version of MS (see P7 L16-20).

12. p7, l9: "Genesis location in ocean surface temperatures less than a user-definable value .." this sentence is not well formulated We have rephrased the sentence following the reviewer's comment (see P7 L12).

13. p7, l23: "The KDE that is sampled are constructed for each grid point based on input data within a specific search range." this sencence does not read well We have rephrased the sentence following the reviewer's comment (see P7 L29).

14. p8, l25: "... not completely similar to the historical ..." maybe this could also depend on the way the termination in historical data is defined? Do all the agencies define the TC termination in the same way? For historical data, termination is defined as the last point of the TC track. TCWiSE can be run purely on historical termination which will result in almost an identical synthetic termination probability compared to historical. However, in TCWiSE it is also possible to add environmental factors to impose TC termination (e.g. wind speed or SST). This is the main source of deviations between synthetic and historical termination.

15. p8, l30 & p9, l3: see my previous comment for p4, l17 See our reaction to specific comment #5 above.

16. p9, l11: wind swaths: it is still a bit unclear what the wind swaths are and how you do generate them - on what variable is the GPD fitted? See our reaction to specific comment #10 above.

17. p11, l30: "MAE": you mean, the MAE between historical and TCWiSE cdf? Please clarify We have added the definitions of MAE and nMAE (see P11 L29-33).

18. p11, paragraph 3.3.1: I would suggest adding formulas with the error indicators used See our reaction to specific comment #17 above.

[Figure]

19. p12, l15: "the genesis patterns ..." this sentence does not read well figure 4: "is the maximum wind speed per TC and not the same as the wind field and/or wind swaths" again, the difference should be explained We have rephrased the sentence about genesis patterns (see P12 L14-15). Moreover, we have added an explanation of the difference between the intensity of the TC eye and wind swaths.

20. p15, l9: "for example" looks out of context Removed. (see P15 L9).

21. p15, l9: the TCWiSE bias of c vs historical looks generally slightly negative. Is it only in this case? Based on our experience on the use of TCWiSE in the Western Pacific Ocean and North + South Indian Ocean, there are no clear biases in terms of the forward speed that are always either positive or negative. The only tendency we noticed is an overestimation of wind speeds at land stations. This is arguably due to the lack of roughnes effects, with the synthetic tracks being largely above water conditions.

22. figure 7: the scale looks in radiants rather than in deg Thank you for noticing this. We have changed this figure in the current version of the MS.

23. figure 8: the bias in TCWiSE max wind looks slightly positive. Is it a systematic tendency or is it random? See our reaction to specific comment #21 above.

24. p19, track termination. To what extend may these differences depend on the uncertainty of historical data on track termination. Differences in track termination between historical and synthetic tracks are compounded over the duration of the simulation. This means that uncertainty in TC track also is partly responsible for the error in the track termination (see P19 L2-14)

25. p20, l7: estimates *of* TC winds We have changed this in the manuscripts, thanks for noticing.

26. p21, l2: How many TC were used for the estimation of the extremes on the historical? How do you ensure the extremes on the historical are compatible with the ones on the synthetic tracks? We have used a total 10 000 years of synthetic TCs in the ex-

treme value analysis. In particular, per grid cell, we have saved maximum wind speeds per TC. Subsequently, using a peak over threshold (POT) method selected a limited number of peaks to fit the Generalised Pareto Distribution (GPD). The historical data are used to create the synthetic tracks which ensure that both are compatible (see P21 L1-3).

27. p21, l5-9: the authors should mention here that the large differences are due to the differences between the historical, used to fit the model, and the observations. They should also mention, earlier, that the historical data are model data, and not observations We have changed the wording slightly in order to emphasize this point (see P22 L2-5).

28. p24, l10: "this makes TCWiSE also more sensitive to input errors compared ...", unclear why this should be: the algorithm used by other authors may as well be sensitive to input errors The reviewer is correct. The point that we are trying to put across is that because TCWiSE is relatively user-friendly, compared to pre-generated global synthetic TC databases, there are more steps involved and therefore room for more user errors.

29. p25, l2: ".. using datasets derived by global climate models .." you mean, CMIPX? How? These models are quite unable to represent properly the TCs. That's a reason why statistical tools like this can come in handy Although TCWiSE can accurately generate high-resolution wind fields it depends on other sources for the definition/determination of the TC genesis, propagation, intensity, and frequency distributions. What we are here stating is that one can use data from Global Climate Model (GCM), such as are being generated by NCAR and GFDL, to infer changes in TC patterns (as done by Knutson et al., 2010) and use these as input. We agree that GCM models often lack resolution and that statistical tools could also be used (e.g. in conjunction with GCM model) to more accurately project changes in the TC distributions. All this together would lead to the generation of more accurate extreme wind field projections by TCWiSE.

[Figure]

30. p25, l20-22: this sentence is a bit unclear and full of repetitions We have changed the wording of this sentence (see P25 L22-24).

31. p25, l25: " It does seem however that synthetic TC tracks have a less clear southwest-"from what do you see this? We have revised and deleted this sentence.

32. p25, l27: subtitute Jetstream with "climate dynamics" We have changed this in the manuscripts, thanks for noticing.

33. p25, l30: "These differences can be attributed to the fact that TC termination can get triggered by ..." this sentence does not read well We have changed the wording of this sentence (see P26 L1).

34. If the tool is open source, I believe it would be useful to provide a link to a code repository After acceptance for publication, we will make the source code publicly available.

---

## Author Comment (AC2) · 28 Oct 2020

Dear editor, dear reviewers,

On the July 31, 2020, we have submitted the following manuscript to the Journal of Natural Hazards and Earth System Sciences titled: "Simulating Synthetic Tropical Cyclone Tracks for Statistically Reliable Wind and Pressure Estimations" (MS No.: nhess-2020-250). On the October 7, 2020, we were informed that the open discussion was completed. In total, we received comments by two reviewers which provided a very positive feedback on the work done and valid suggestions. We would like to acknowledge their time and efforts, which have led to an improvement in the quality of our

manuscript. Below you find a point-by-point reply to all specific questions and suggestions. Attached you also find the revised manuscript with the changes made to address the review comments tracked.

Kind regards, Kees Nederhoff

—

James Done (Referee #2)

Specific Main Comments:

1. I agree that synthetic track simulation adds events and overcomes the sampling problem. But these need to be interpreted in the correct context. These synthetic tracks are constrained to reproduce the statistics of the historical record. This means that this tool would not, for example, produce a Hurricane Sandy-like track before Sandy occurred in the historical record. A physical model on the other hand has the potential to produce physically credible but not observed track behaviors. I suggest making this point in the discussion. We agree and have added the comment of physically-credible and statistically-unlikely tracks in the discussion (P24 L18-19). Note that because we divide the basins into grid cells and define the distributions per cell, the tool is capable of generating tracks that have not occurred before.

2. Another limitation of the tool is the assumption of stationarity in the historical record. We know that change has been detected in some TC characteristics in some regions (Knutson et al. 2019). Perhaps this limitation can be stated in the discussion. We have added the assumption of stationarity to the discussion (P24 L19).

3. I'm glad to see the option to include inland wind decay of Kaplan and De Maria 1995) in addition to the implicit decay through the KDE of Vmax. But it's important to state in the manuscript that at-sea winds will still extend inland before the TC center crosses the coast and the Kaplan and De Maria wind decay turns on. I think this is a possible reason for your high bias in 10-year return winds in some coastal regions (Fig.

10). We extended our discussion on the limitations of the landward decay based on De Maria and Kaplan in the discussion section (P25 L9-10).

4. Section 2.5: Can you explain in more detail how asymmetry is considered? The Vmax in BTD is ground-relative and so includes a component of asymmetry. Did you remove the component of asymmetry from the BTD Vmax before creating th synthetic tracks and running the symmetric Holland model (and then add asymmetry back to the spatial wind field afterwards)? TCWiSE in its generation of synthetic tracks does not take into account asymmetry since it is focused on intensity evaluation of the eye. The Wind Enhanced Scheme (WES; Deltares, 2018) handles asymmetry in the computation of the spatially-varying winds. In particular, asymmetry is removed from the synthetic track via the relationship of Schwerdt (1979) prior to fitting the Holland wind model and asymmetry is added afterward again on the spatial wind fields. In this paper, we mainly want to focus on the TCWiSE tool and have therefore not added this explanation, however, we did add a more explicit reference about WES (see P9 L18).

5. There are a couple of notable omissions from the reference list. Arthur (in review) has a paper under discussion at NHESS that describes a synthetic track model that has similar functionality to this study. Lee et al. (2018) published a synthetic track model that differs from your data-driven approach by accounting for environmental drivers of TC behavior. Thank you for pointing this out, we have included these references in the introduction (P3 L1-2 and P3 L8-9).

6. The paper is generally well written. But there are a few grammatical quirks and awkward word choices that can be corrected by a thorough review of English grammar. To improve the language in the MS, a native speaker has reviewed the paper which hopefully resolved the few grammatical quirks and awkward word choices.

Specific comments

1. Abstract, lines 10-12: This sentence makes the point that short historical records may not represent the parent population. This is a valid point but I suggest not using

the term 'future TCs' in this sentence because that implies climate change and nonstationarity which is a separate issue. We agree with Reviewer 2 and have changed it into 'hypothetical'.

2. Introduction: The sentence spanning lines 27-29 about first and second order effects doesn't appear to fit well in this paragraph about extreme value modeling. We have moved this part of the introduction to the first paragraph since this is a point we would like to make.

3. The introduction talks a lot about the multi-hazard nature of TCs but then the paper describes a tool for TC wind only. I suggest toning done the discussion of surge, waves and rainfall in the discussion and just mentioning it briefly. We agree and have therefore changed the wording in the introduction (see P1 L22).

4. To improve the flow of the introduction, can the point about the need for a larger sample size be made just once? It is currently discussed twice in the first and third paragraphs. We have removed the first reference of the sample size in order to improve the flow of the introduction.

5. Section 2.2. Why not choose a threshold of 17ms to include all Tropical Storms rather than is seemingly arbitrary 25 m/s? We agree and have changed this value in the default settings (the user can always change it to another value).

6. Section 2.4.3. What are the units '10kn/s'. Do you mean knots? Yes, we have changed kn/s to knots.

7. Can you comment on the computational performance of the tool? How long does it take to run 10,000 years of the North Atlantic, for example? Computational performance depends strongly on numerical settings and computational power available to the user. We did add in a separate discussion section in which we mention the computational cost for the configuration we ran in our case to give readers an impression (see P26 L18-21).

8. Fig. 5. Would it be useful to additionally plot the difference field to highlight the differences discussed in the main text? Figure 5 (and subsequent ones) are used to show qualitative patterns that are supported with quantitative evaluation via metrics such as the Kirchhofer metric score. We feel that a difference plot will focus too much on the minor variation between historical and synthetic spatial maps instead of focusing on the larger-scale patterns.

9. In Section 3.3, use of the fourth-highest recorded value for the 10-year return wind will probably be noisy. Would it be better to fit an extreme value distribution to the observations to estimate the return value? This may produce a better agreement with the model. We agree that there are uncertainties associated with this empirical approach and that an extreme value distribution might give less noisy results. This is why we show both the non-parametric (empirical) and parametric (fit) results from TCWiSE.

10. Figure 11: Can you clarify what the historical TC wind data are please? Is it Holland model run along historical tracks? Yes, tracks and intensity. The historical is simply the BTD in combination with the Holland wind profile to get to spatially-varying wind fields. The synthetic TC is the same but then based on synthetic tracks and intensities. In order to clarify this, we have changed the caption of the figure to include these explanations (P22 L2-5).

11. The description of Fig. 11 in main text has 'Port Arkansas'. The correct name is Port Aransas. 12) Figures 10 and 12: Please state the grid spacing used in these Thank you for noticing. We have corrected the text and added the spacing (P21 L9-10).

12. Section 4.2. I don't see what you are referring to about the synthetic TC tracks having a less clear southwest to northeast orientation. I think this needs to be quantified in some way or excluded from the manuscript. We have deleted this sentence.

13. The Hoek (2017) reference was incomplete in my pdf version. Thank you for noticing and it has been corrected (P29 L4-5).

14. The Bader (2019) reference is missing from the reference list. Thank you for noticing and it has been added (P28 L5-6).

---

## Author Comment (AC3) · 28 Oct 2020

See attached

Please also note the supplement to this comment:
https://nhess.copernicus.org/preprints/nhess-2020-250/nhess-2020-250-AC3-supplement.zip

---

## Author Comment (AC4) · 28 Oct 2020

See attached

Please also note the supplement to this comment:
https://nhess.copernicus.org/preprints/nhess-2020-250/nhess-2020-250-AC4-supplement.zip
* * *

---

## Author Response (AR2)

**Rebuttal letter manuscript "Simulating Synthetic Tropical Cyclone Tracks for Statistically Reliable Wind and Pressure Estimations"**

Dear editor, dear reviewers,

On July 31, 2020, we have submitted the following manuscript to the Journal of Natural Hazards and Earth System Sciences titled: "Simulating Synthetic Tropical Cyclone Tracks for Statistically Reliable Wind and Pressure Estimations" (MS No.: nhess-2020-250). On October 7, 2020, we were informed that the first round of open discussion was completed. We resubmitted the revised manuscript on October 29, 2020 after recieving comments by two reviewers which provided a very positive feedback on the work done and valid suggestions. On January 14, 2020, we were informed that the second round of open discussion was completed. We received a few minor comments by one reviewer and the acceptance by the second reviewer. Below you find a point-by-point reply to all specific questions and suggestions. Attached you also find the revised manuscript with the changes made to address the review comments tracked.

Kind regards,

Kees Nederhoff
* * *
**Referee #1: Mentaschi, Lorenzo**

General Comments:

- Comment 5: Maybe a data-driven way to consider SST could be using a joint distribution space-SST for the genesis?

We agree with Reviewer #1. A joint probability map as function of SST, could be a possible way to estimate how changes in climate, via SST, affect TC generation. We have added this to the discussion section on P25-L3.

- Comment 7: Then a further quesion arises: the SST used as an input spans over e few decades, while TCWiSE can generate TC data over thousands of years. How are the years of the input SST matched with the ones of the similation? This should be clarified in the manuscript.

In each TCWiSE application, either based on current climate or climate projections, stationarity is assume. In other words, the cyclone and SST characteristics are not expected to evolve in time. The long periods for which data are generated, for instance 1,000 years are not to be seen a forecasts for such periods but the generation of low probability events, as the 1,000-year event of the current climate. See also the discussion section P24-L20.

- Comment 10: What the authors write is clear, but IMO the differences between wind maxima, wind swath and wind maps should be clarified further inside the manuscript, as now the reader has to guess: the wind swath is used the first time at P5 L20 without introducing it, and in the label of figure 4 it is stated that "the maximum sustained wind speed is the maximum wind speed per TC and not the same as the wind field and/or wind swaths"

We understand the confusion of Reviewer #1 and therefore went over the MS and clarified the language. Please see the revised MS and for example the caption of Figure 4.
- Maximum sustained wind speed is the intensity of the eye.
- Time and space varying surface winds refer to the time-varying 10 m level (surface) wind fields (three-dimentional wind velocities). With field being used to describe spacevarying wind velocities (two-dimentional wind velocities).These are also referred to solely as (surface) wind fields in the MS.
- Wind swaths are maxima per TC (i.e. by computing at a certain location maximum in time of the wind velocity) and can also be associated to given probabilities (e.g. wind swath with a return period of 100 years). Wind swaths are also called wind extremes in the MS.

**James Done (Referee #2)**

All comments by James Done were addressed in the first round of revisions.